# Neustrelitz Total Electron Content Model for Galileo Performance: A Position Domain Analysis

**DOI:** 10.3390/s23073766

**Published:** 2023-04-06

**Authors:** Ciro Gioia, Antonio Angrisano, Salvatore Gaglione

**Affiliations:** 1Independent Researcher, 21020 Brebbia, Italy; 2Department of Engineering, Messina University, 98100 Messina, Italy; 3Department of Science and Technology, Parthenope University of Naples, 80143 Napoli, Italy

**Keywords:** NTCM, NeQuick-G, Klobuchar, Ionosperic model

## Abstract

Ionospheric error is one of the largest errors affecting global navigation satellite system (GNSS) users in open-sky conditions. This error can be mitigated using different approaches including dual-frequency measurements and corrections from augmentation systems. Although the adoption of multi-frequency devices has increased in recent years, most GNSS devices are still single-frequency standalone receivers. For these devices, the most used approach to correct ionospheric delays is to rely on a model. Recently, the empirical model Neustrelitz Total Electron Content Model for Galileo (NTCM-G) has been proposed as an alternative to Klobuchar and NeQuick-G (currently adopted by GPS and Galileo, respectively). While the latter outperforms the Klobuchar model, it requires a significantly higher computational load, which can limit its exploitation in some market segments. NTCM-G has a performance close to that of NeQuick-G and it shares with Klobuchar the limited computation load; the adoption of this model is emerging as a trade-off between performance and complexity. The performance of the three algorithms is assessed in the position domain using data for different geomagnetic locations and different solar activities and their execution time is also analysed. From the test results, it has emerged that in low- and medium-solar-activity conditions, NTCM-G provides slightly better performance, while NeQuick-G has better performance with intense solar activity. The NTCM-G computational load is significantly lower with respect to that of NeQuick-G and is comparable with that of Klobuchar.

## 1. Introduction

The Sun emits electromagnetic radiation ionizing different atmospheric layers and inducing the Earth’s magnetic field variations. In addition, ultraviolet radiations convert the neutral particles in the Earth’s atmosphere to electrons and ions. Overall, these electrons are concentrated in the altitude range from ∼60 km to 1000 km; however, their characteristics are spatially and temporally dynamic. For low frequency signals, this effect is exploited to enable long distance communications, while at much higher frequencies, radio waves pass right through the ionosphere; however, the speed of the signal is influenced by the density of electrons. This phenomenon strongly affects several applications including global navigation satellite system (GNSS)-based navigation. The ionosphere has attracted a large interest of the GNSS community in recent years because the last solar peak was observed in 2013 and a new one is approaching. The high solar activity could severely affect GNSS positioning because the ionosphere is one of the most significant error source for users in open-sky conditions and it can vary between a few and tens of meters [1]. In order to have an accurate position solution, the ionospheric effect needs to be mitigated; different approaches can be used depending on the receiver characteristics. In particular, the ionospheric error can be reduced using multi-frequency measurements, exploiting corrections provided by an augmentation system, and relying on a model [2,3]. In the first approach, the dispersive nature of the ionosphere is exploited and its effect is mitigated using a linear combination of measurements from two frequencies [4]. Although this approach allows the removal of up to 99% of the ionospheric error, it requires the use of a multi-frequency receiver that is in general more expensive than a single frequency device; in addition, the combination of the measurements leads to an increased measurement noise [5]. Single frequency devices could limit the ionospheric error using an augmentation system such as the European Geostationary Navigation Overlay Service (EGNOS) [6] or Wide Area Augmentation System (WAAS) [7]; these systems broadcast ionospheric correction for specific coverage areas and the performance of these systems has been analysed in [8]. The satellite-based augmentation system (SBAS) service coverage is strongly impacted by the ground reference stations network for monitoring satellites signals and computing ionospheric corrections [9]. With this approach, a receiver needs to access the correction through an internet connection (e.g., for EGNOS [10]) or processing the SBAS signals, adding complexity to the device and to the processing chain. The ionospheric error can be removed using differential techniques such as differential GNSS (DGNSS); in this case, the relative positioning techniques exploit the spatial correlation of the ionospheric. Another example is the usage of real time kinematic (RTK) methods; in [11], the authors demonstrated the integration of local ionospheric and tropospheric corrections as constrain to the precise point positioning estimation. In [12], the introduction of the global ionospheric map in RTK positioning was investigated. Single frequency stand-alone devices need to mitigate the ionospheric error using a model. Recently, the empirical model Neustrelitz Total Electron Content Model for Galileo (NTCM G) [13] has been proposed as an alternative to Klobuchar [14] and NeQuick-G [15] (currently adopted by GPS and Galileo, respectively). The three models exploit different approaches and have different characteristics: the Klobuchar model is quite simple to implement and it is able to correct up to 50% of the ionospheric error; some advantages have been demonstrated when using the NeQuick-G model, which is able to correct up to 70% of the ionospheric error [15], though it has an higher computation load. NTCM-G is an empirical model that has a performance similar to that of NeQuick-G but with a lower complexity. Several studies have evaluated the performance of ionospheric correction models. The performances of NeQuick-G and Klobuchar were compared in [16,17]; from the analysis, it emerged that NeQuick-G performed better than the Klobuchar model. The performance of the two models were also analysed in non-nominal conditions during St. Patrick’s storm in [18]; the study concluded that a larger degradation of the horizontal and vertical error was observed when the Klobuchar model was used. The performance of the NTCM-G model has been compared with respect to that of NeQuick-G and Klobuchar in [19]; the study showed that, considering the recently increasing solar activity, NTCM-G corrections presented accuracy comparable with that of the NeQuick-G model and better than that of the Klobuchar one. The performance of the application of the NTCM-G model in the position domain has been investigated in [20]: from the analysis, it emerged that the NTCM-G model has a performance similar to that of NeQuick-G; the assessment was carried out considering only two solar activity conditions.

In this work, we assess the performance of the three models considering different solar activities. Specifically, four days have been considered: one with intense solar activity (in September 2014), one with medium solar activity (in March 2016), one with low solar activity (December 2017) and one with a rising solar activity (in January 2023). A more detailed description of the dataset used is provided in Section 3. In addition to the solar activity, the geomagnetic field strongly impacts the ionospheric delay. To evaluate the performance of the model in different geomagnetic conditions, three different stations are considered. One station is in the proximity of the geomagnetic equator, one at middle geomagnetic latitude and one at high geomagnetic latitude leading to three different geomagnetic conditions. Finally, in addition to the previous works, the computational load of the three models was also compared. Such analyses allow a user to properly identify which model can be used according to its computational capacity, solar activity, geomagnetic conditions, and the accuracy required by the application.

The remainder of the paper is structured as follows. In Section 2, the navigation solution algorithm and the ionospheric models are presented. Section 3 describes the experimental set-up and the data analysed for the performance assessment; in Section 4, the results are discussed. Finally, Section 5 concludes the paper.

## 2. Absolute Positioning Computation and Ionospheric Models

In this section, the overall approach adopted for the evaluation of the absolute positioning is briefly introduced along with the description of the three ionospheric models used.

### 2.1. Absolute Positioning

In single point positioning (SPP) mode, a GNSS receiver computes its absolute position processing measurements of at least four space vehicles (SVs) from the same constellation. This method is based on pseudorange observations (PR), the equation for which is as follows:(1)$$PR_{i}=\rho_{i}-\Delta T_{i}\pm Rel_{i}+I_{i}+T_{i}\mp b+\varepsilon_{PR}$$
where $\rho_{i}$ is the geometric range between the receiver and *i*^th^ satellite, b is the receiver clock offset in distance unit, $\Delta T_{i}$ is the *i*^th^ satellite clock error, $Rel_{i}$ is the relativistic error, $I_{i}$ is the ionospheric error, $T_{i}$ is the tropospheric delay, and the term $\varepsilon_{PR}$ includes all the remaining errors not corrected by the models (multipath, receiver noise). The unknown parameters of the problem are the receiver coordinates (embedded in $\rho_{i}$) and the receiver clock offset b.

Equation (1) is partially corrected for satellite clock, relativistic, ionospheric, tropospheric errors. The satellite clock error is corrected using a polynomial model whose parameters are broadcast in the navigation message [21]. The relativistic effects are corrected using the models reported in [3,5]. Tropospheric effect is limited using the Saastamoinen model [22]. Ionospheric errors are corrected using one of the three models considered in this work. The corrected pseudoranges are then linearized around a set of approximate unknown parameters in order to compute the receiver position. A set of *m* pseudorange linearized equations has to be processed to obtain a measurement model, the matrix form of which is as follows:(2)$$\underline{z}=H\underline{∆x}+\underline{\varepsilon }$$
where $\underline{z}$ is the measurements vector, containing the difference between actual and computed pseudoranges, H is the design matrix, $\underline{\varepsilon }$ contains the un-modelled and the residual errors, and $\underline{∆x}$ is the correction vector to be applied to the approximate receiver coordinates and clock offset. $\underline{∆x}$ must be computed by an estimation technique.

The position estimation is usually performed using a recursive algorithm based on a weighted least squares (WLS) method or Kalman filter (KF); in this work, a WLS approach is used with weights based on the satellites’ elevation.

The main inputs of SPP, as shown in Figure 1, are GNSS observables, and ephemerides. A suitable algorithm (in the “orbital propagator” block) computes the satellite positions and satellite clock errors at the transmission epochs. The measured pseudoranges are corrected for the tropospheric error, relativistic and satellite clock errors; using appropriate models, the ionospheric corrections are computed in the green block using one of the three ionospheric models considered in this work. Finally, the corrected measurements are processed using the WLS estimator.

### 2.2. Ionosperic Models

The ionosphere is a dispersive medium, located from 70 to 1000 km above the Earth’s surface; in this region of the atmosphere, ionised electrons behave as free particles, influencing GNSS signals and therefore all the measurements. In particular, a delay is introduced on the code measurements, while an advance in carrier phase measurements is observed. The ionospheric delay for the pseudorange measurements is as follows:(3)$$I=\frac{40.3*10^{16}}{f^{2}}*TEC$$
where f is the signal frequency in Hz, TEC is the total electron content in electrons/m^2^ computed along the receiver–satellite path.

For single frequency stand-alone receivers, this delay is estimated using a model; the three models considered in this work are briefly described in the following sections.

#### 2.2.1. Klobuchar

The ionospheric Klobuchar model, as a result of its simplicity and low computational load, is widely used by single frequency GNSS receivers to correct approximately 50% of the ionospheric error, as described in [14]. The model design is based on the Bent model [23], and uses the single layer approach assuming the ionosphere concentrated in an infinitesimal layer at approximately 350 km from the Earth’s surface. The model estimates the ionospheric delay in exploiting eight coefficients broadcast in the GPS navigation message [21]. Two different estimations are provided for day and night time.

#### 2.2.2. NeQuick-G

Three versions of the NeQuick model have been developed. The first version, known as NeQuick, was developed by the Abdus Salam International Center of Theoretical Physics (ICTP) and the University of Graz [24]. It is a semi-empirical model that describes spatial and temporal variations of the electron density, and it is based on the Di Giovanni and Radicella (DGR) ionospheric profiler [25]. The adaptation of NeQuick for the Galileo single frequency ionospheric correction algorithm (NeQuick-G) has been performed by the European Space Agency (ESA); the algorithm description and a step-by-step implementation are provided in [15]. NeQuick-G estimates total electron content (TEC) using three parameters broadcast in the Galileo navigation message [26]. Finally, the last version of NeQuick is identified as NeQuick2 [27]; with respect to the previous versions, NeQuick2 is characterized by a new formulation of the topside representation exploiting topside soundings data from the ISIS-2 (International Satellites for Ionospheric Studies) [28]. In this work, we used the NeQuick-G model; specifically, the software implementation developed by the European Commission Joint Research Centre (JRC) available at https://www.gsc-europa.eu/support-to-developers/ionospheric-correction-algorithms/nequick-g-source-code (accessed on 8 August 2022) has been used. The performance of the algorithm was analysed in [29], where the author reported that the algorithm is faster than the ESA version and provides a more accurate TECs estimate.

#### 2.2.3. NTCM-G

NTCM is an empirical model developed by Deutsches Zentrum für Luft- und Raumfahrt (DLR) [30]; it is a single layer TEC model similar to Klobuchar. Several modifications of the model have been proposed [19,20,31,32]; however, all the versions share a common approach: TECs are estimated as a function of geophysical parameters such as the geographic location, time and solar activity. The version used in this work is identified as NTCM-G and is the one described in [13]; the model has been implemented in Matlab and the module has been included in a customized navigation algorithm.

A block diagram with the main elements of the implemented version is shown in Figure 2. The TEC is computed as the products of five different contributors:(4)$${VTEC}_{NTCM-G}=F_{1}*F_{2}*F_{3}*F_{4}*F_{5}$$
where $F_{1}$ takes into account the local time dependency, $F_{2}$ represents the seasonal variation, $F_{3}$ models the effect of the geomagnetic field and it is driven by the geomagnetic latitude, $F_{4}$ considers the Appleton anomaly [33,34], and $F_{5}$ takes into account the solar activity and is based on the three ionospheric parameters broadcast in the Galileo navigation message.

The solar activity dependence of the ionospheric delay is taken into account by the *F*_5_ factor computed as follows:(5)$$F_{5}=k_{11}+k_{12}\cdot Eff_{IONI}$$
where $k_{11}$ and $k_{12}$ are two coefficients set to 1.41808 [TECU] and 0.13985 [TECU/sfu], respectively. $Eff_{IONI}$ is the effective ionization level determined as follows:(6)$$Eff_{IONI}=\left|\sqrt{\left(a_{0}^{2}+1633.33\cdot a_{1}^{2}+4802000\cdot a_{2}^{2}+3266.67\cdot a_{0}\cdot a_{2}\right)}\right|$$

$a_{0}$, $a_{1}$ and $a_{2}$ are the three effective ionisation level coefficients broadcast in the Galileo navigation message. The effective ionization level is measured in solar flus unit (sfu).

A complete description of the different factors and the related formulas is available in [13,20,30,35,36,37].

## 3. Experimental Setup

As mentioned in the previous sections, ionospheric delay is influenced by several elements including geomagnetic conditions, local time, and solar activity. Different criteria could be used to select the stations to be analysed, for example, the stations with largest TEC values observed in the global ionospheric maps (GIMs) or considering the largest variation of the TEC. In this work, to analyse the impact of the geomagnetic latitude on the considered models, three stations at different geomagnetic latitudes have been selected: an equatorial station in Arequipa, a middle latitude station in Madrid and a high latitude station in Baker Lake. The coordinates of the stations are reported in Table 1, while their geographical distribution is shown in Figure 3.

To evaluate the performance in different solar activity conditions, four days of data are processed; each selected day is characterized by solar activity. As a high solar activity case, 29 September 2014 has been considered; it is the day with the highest effective ionization value (179.12) since the end of 2014. For medium solar activity conditions, the 26 March 2016 has been selected; it is characterized by an average value of the effective ionization (68.52) of the considered period as shown in Figure 4. For low solar activity case, 15 December 2017 has been considered, being characterized by the lowest value of effective ionization (29.18). Finally, considering the increased solar activity, a more recent day has been analysed as well; the selected day is 12 January 2023, with an effective ionization of approximately 141. The solar activity conditions and the selected date are reported in Table 2.

## 4. Results

In order to assess the performance of the ionospheric models considered, the following metrics are used:-Execution time, defined as the time needed to compute the ionospheric corrections;-Position error, computed as the difference between the estimated position solution and the reference coordinates of the station. The parameters used to evaluate the performance are the mean, standard deviation and 95th percentile of horizontal and vertical errors.

In Figure 5, the execution times of the three algorithms are taken into account for a single day (DOY 12, 2023) and for a single station (Arequipa). The execution times for other days and stations are very similar; therefore, they are omitted to avoid repetition of similar results. In particular, the figure represents the ratio between the execution times of couples of models; for instance, the blue line is the ratio between the execution time of NeQuick-G and Klobuchar. The *y*-axis is in logarithmic scale. Such an approach has been adopted because the represented values were extremely different between NeQuick-G and the other two models. Indeed, the NeQuick-G execution time is approximately 48 times larger than that of NTCM-G and approximately 58 times larger than that of Klobuchar. The execution times of NTCM-G and Klobuchar are very similar, as demonstrated by the yellow line.

In the Figure 6, Figure 7, Figure 8 and Figure 9, the horizontal errors, obtained with the three models, are shown. In addition to epoch-by-epoch errors (represented by blue dots), for each model, a circle containing 95% of the dots is drawn; this should facilitate the comparison between the configurations. The figures show the results of the ionospheric models in rows:-the upper boxes concern the results obtained using the Klobuchar model;-the central boxes concern results obtained using NeQuick-G;-the lower boxes concern results obtained using the NTCM-G model.

The errors attained in case of intense solar activity are shown in Figure 6, where the NeQuick-G model exhibits lower errors at Madrid and Arequipa, and larger errors at Baker Lake, while the results obtained using NTCM-G and Klobuchar are very similar. A more recent day with a high solar activity is considered in Figure 9; in this case, NTCM-G provides an improved performance with respect to those of both the NeQuick-G and the Klobuchar models for all the stations.

With medium solar activity conditions, the differences among the three models are very limited, as shown in Figure 7; in these conditions, NTCM-G seems to outperform NeQuick-G and Klobuchar as well. Figure 8 shows the results for low solar activity: NTCM-G and NeQuick-G practically overlap and they provide a better performance with respect to that of Klobuchar with a larger difference at high latitude.

In Figure 10 and Figure 11, the mean position errors, respectively, horizontal and vertical, are shown. For the days characterized by high solar activity (represented by blue and yellow bars), the errors are generally larger, demonstrating the difficulties of the models as regards working in these conditions. For the station of Arequipa, being the nearest to geomagnetic equator, the errors are also generally larger. Considering the horizontal mean errors, the models’ performances are very similar. NeQuick-G works slightly better with high solar activity, and the results obtained with the NTCM-G and Klobuchar models are very similar. Analogous considerations can be made also for vertical mean errors: the model performance is similar.

In Figure 12 and Figure 13, the standard deviations of the horizontal and vertical position errors are shown, respectively. For both the horizontal and vertical channels, the values related to the days characterized by high solar activity (represented by blue and yellow bars) are generally higher than the other cases. The standard deviation analysis leads to considerations similar to the ones drawn for the mean errors.

In Figure 14 and Figure 15, the cumulative distribution function (CDF) of the position errors, respectively, horizontal and vertical, are shown; the three models are plotted with different colours, and errors with different solar activity are considered in different boxes. On the horizontal plane (Figure 12), it can be clearly observed how the increased solar activity affects the positioning performance as the curves are steeper in the panel on the left. In these conditions, the horizontal error is lower than 1 m with a probability of approximately 70% (72% for NTCM-G, 70% NeQuick-G and 67% for Klobuchar), while in the case of intense solar activity, the probability of having a similar error decreases to less than 40% (38% NeQuick-G, 33% NTCM-G and 25% Klobuchar). In low and medium solar activity conditions (left and central panels), the NTCM-G model has a better performance, while, NeQuick-G has a better performance with intense solar activity (panel on the right).

The CDF of the absolute value of the vertical error is shown in Figure 15. In low and medium solar activity conditions (left and central panel), NTCM-G and NeQuick-G models slightly outperform Klobuchar; with high solar activity (panel on the right), Klobuchar slightly outperforms the other two models.

## 5. Conclusions

In this study, three ionospheric models are considered: Klobuchar, NTCM-G and NeQuick-G. The performance of the models is evaluated in the position domain using a SPP approach; the assessment has been performed including the three models in a customized positioning software. Data of three stations in different geomagnetic locations are used for the assessment. In order to evaluate the impact of the increasing solar activity, four days with low, medium and high solar activity are considered.

From the interpretation of the results, it has emerged that in low and medium solar activity conditions, the NTCM-G model has a better performance, while, NeQuick-G has a better performance with intense solar activity. The impact of the solar activity can also be clearly observed in the positioning error domain: in low and medium solar activity, a horizontal error lower than 1 m was obtained with a probability of approximately 70%, while in the case of intense solar activity, the probability of having a similar error decreased to less than 40%. The performance of the three models is also assessed in terms of execution time. From the analysis, it can be observed that the Klobuchar and NTCM-G models have a similar computational load, while NeQuick-G requires a larger computation effort; thus, its execution time is approximately 50 times higher than that of the other two models.

## Figures and Tables

**Figure 1 sensors-23-03766-f001:**
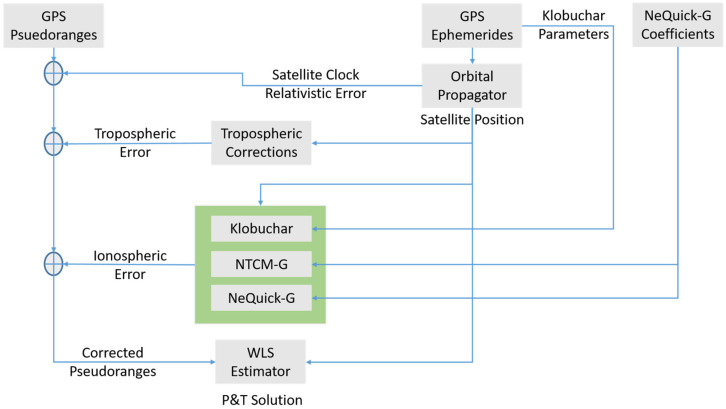
SPP diagram. The green block includes the three considered ionospheric algorithms.

**Figure 2 sensors-23-03766-f002:**
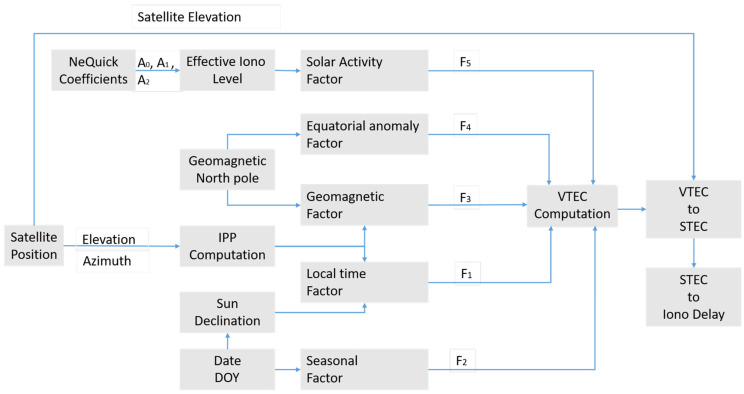
Block diagram of the NTCM-G algorithm implemented in Matlab.

**Figure 3 sensors-23-03766-f003:**
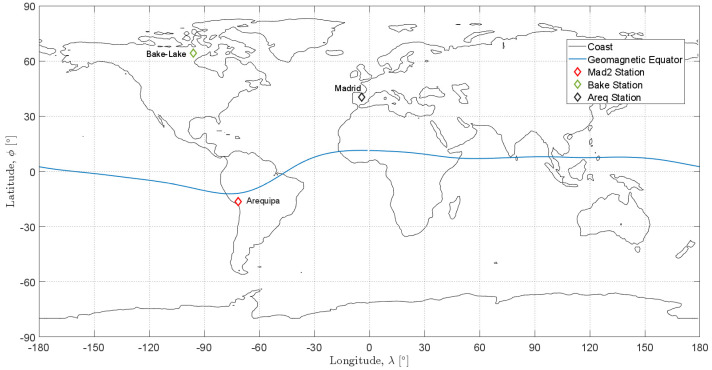
Distribution of the stations used for the analysis.

**Figure 4 sensors-23-03766-f004:**
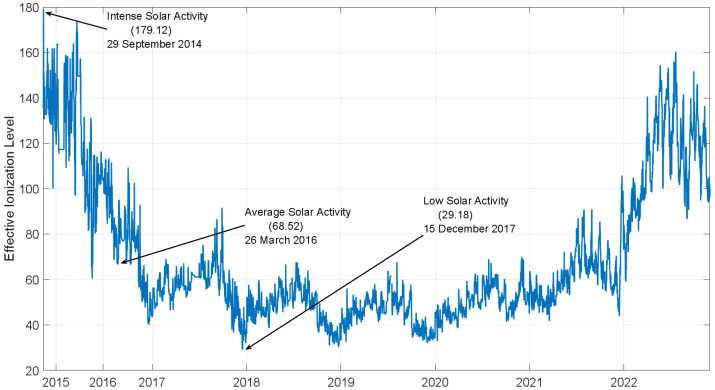
Effective ionization level in the period September 2014–December 2022.

**Figure 5 sensors-23-03766-f005:**
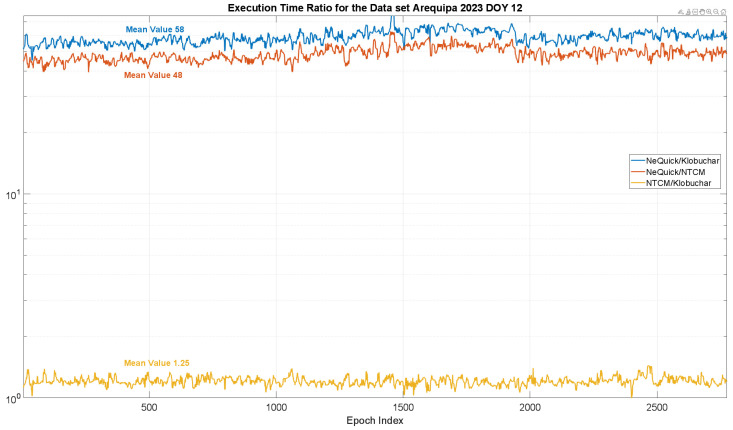
Execution time ratio for the DOY 12, 2023 and for the Arequipa station. The ratio between execution times of couples of models is considered.

**Figure 6 sensors-23-03766-f006:**
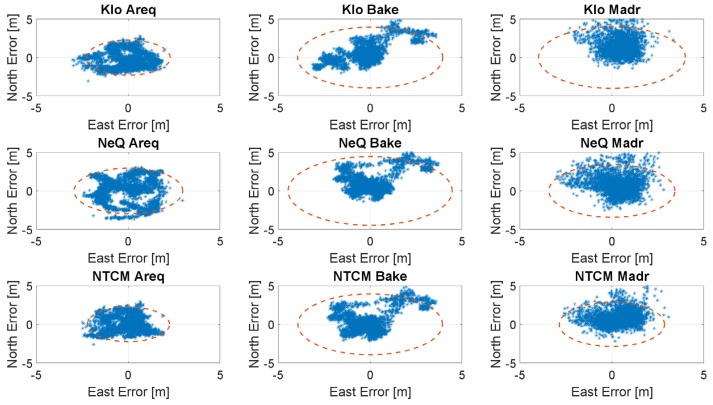
Horizontal errors for the three stations with intense solar activity (29 September 2014).

**Figure 7 sensors-23-03766-f007:**
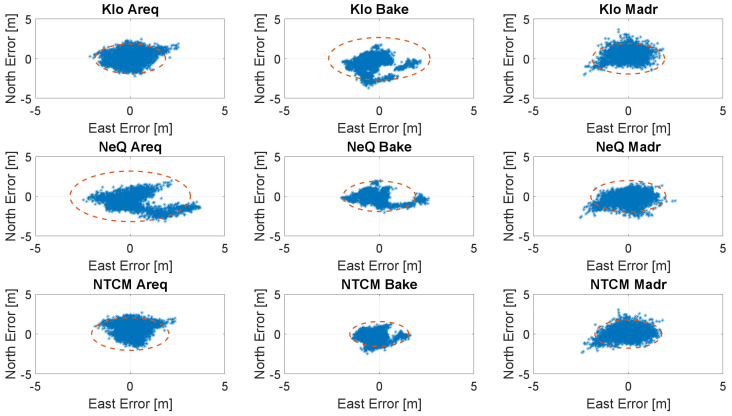
Horizontal errors for the three stations with medium solar activity (26 March 2016).

**Figure 8 sensors-23-03766-f008:**
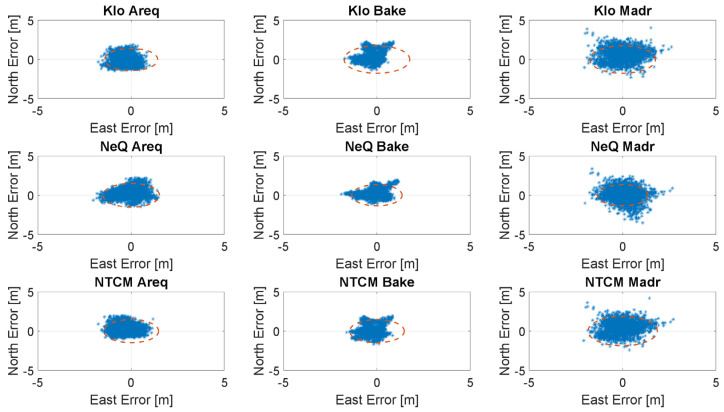
Horizontal errors for the three stations with low solar activity (15 December 2017).

**Figure 9 sensors-23-03766-f009:**
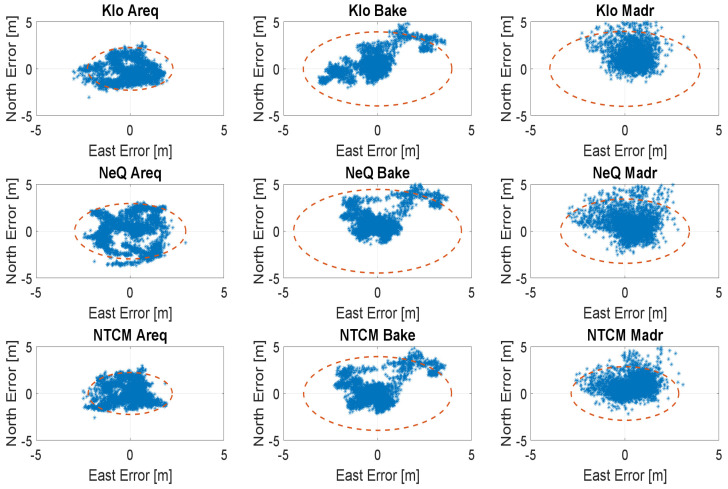
Horizontal errors for the three stations with high solar activity (12 January 2023).

**Figure 10 sensors-23-03766-f010:**
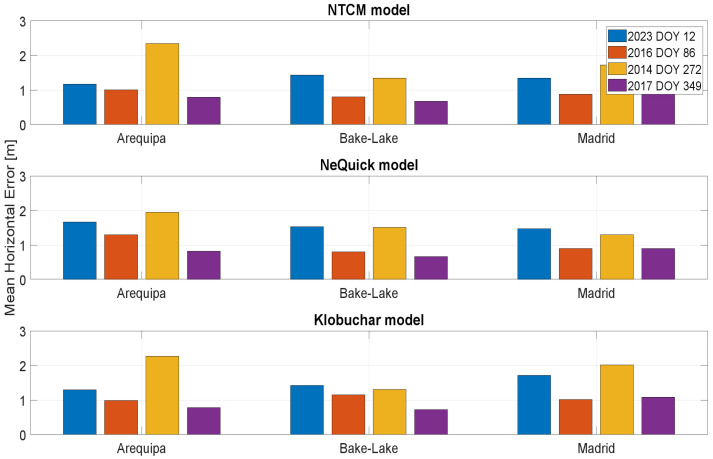
Mean horizontal positioning error for the stations considered in different solar conditions.

**Figure 11 sensors-23-03766-f011:**
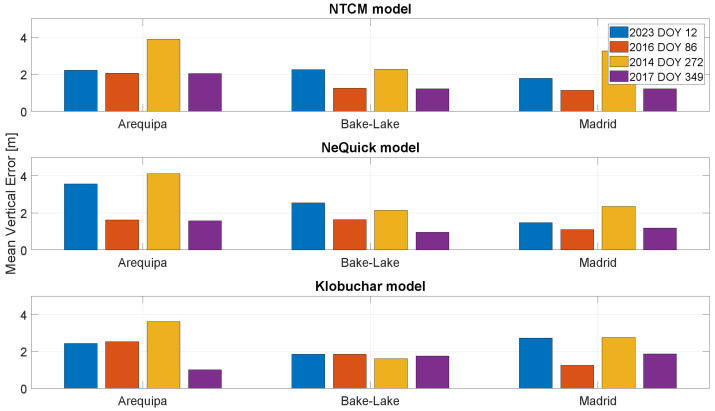
Mean vertical positioning error for the stations considered in different solar conditions.

**Figure 12 sensors-23-03766-f012:**
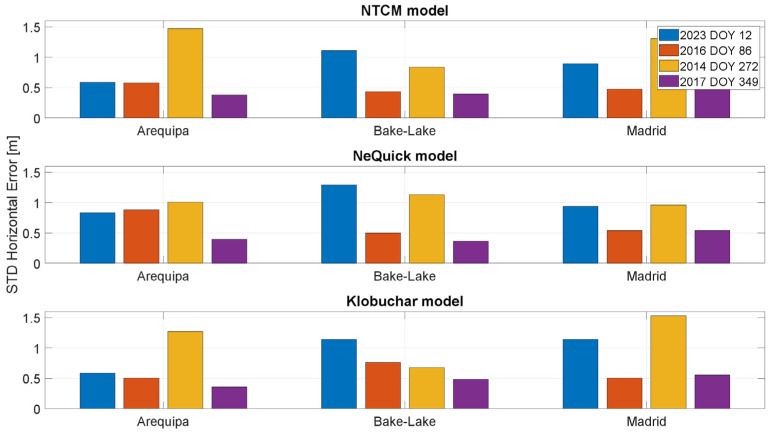
Standard deviation of the horizontal positioning error for the stations considered in different solar conditions.

**Figure 13 sensors-23-03766-f013:**
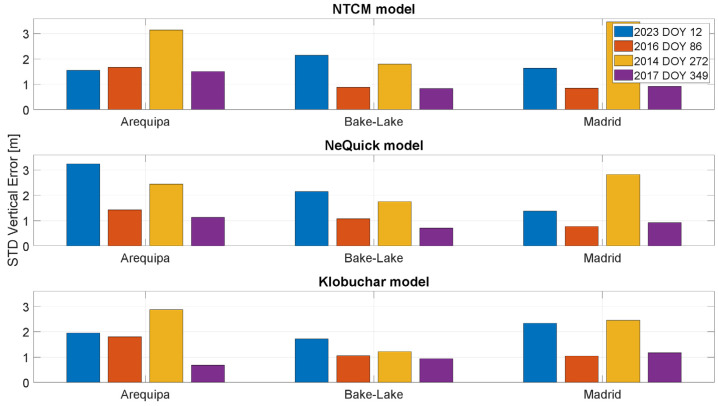
Standard deviation of the vertical positioning error for the stations considered in different solar conditions.

**Figure 14 sensors-23-03766-f014:**
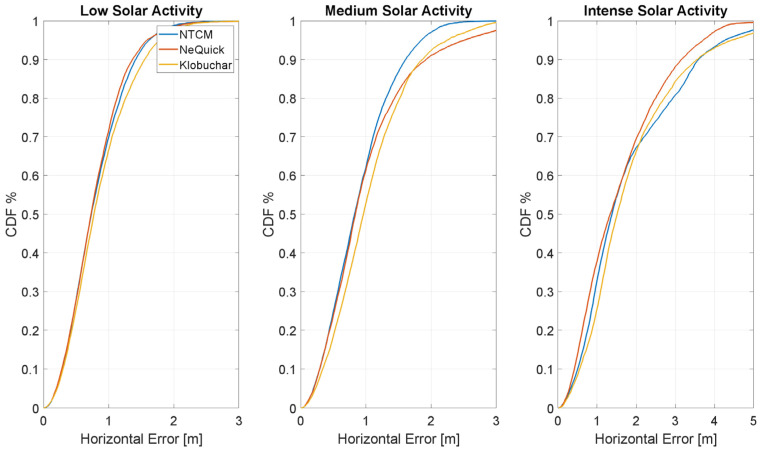
CDF of the horizontal positioning error in different solar conditions.

**Figure 15 sensors-23-03766-f015:**
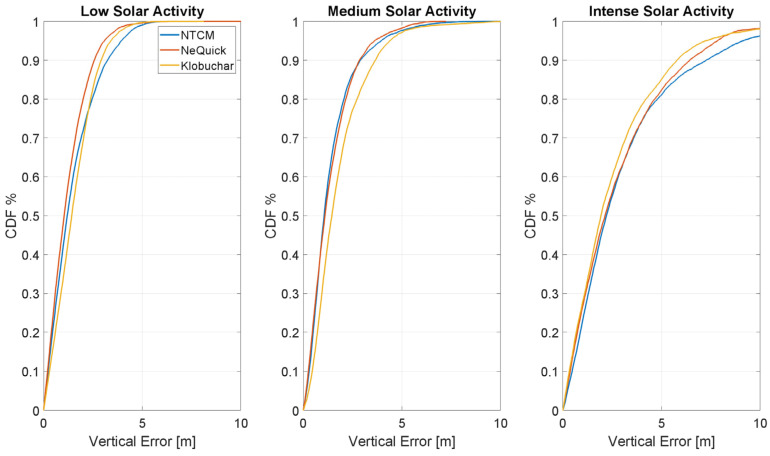
CDF of the vertical positioning error in different solar conditions.

**Table 1 sensors-23-03766-t001:** Coordinates of the stations considered in the analysis.

Station ID	Location	Latitude [Deg]	Longitude [Deg]	Altitude [m]	Geomagnetic Lat [Deg]
AREQ	Arequipa (Peru)	16.4655157 S	71.4927960 W	2488.926	6.21 S
BAKE	Baker Lake (Canada)	64.317820 N	96.0023435 W	4.409	73.18 N
MAD2	Madrid (Spain)	40.429161 N	04.2496593 W	829.456	43.63 N

**Table 2 sensors-23-03766-t002:** Solar flux conditions for the days considered in the analysis.

DOY	Year	Ionization Level	Solar Activity
272	2014	179.12	High
086	2016	68.52	Medium
349	2017	29.18	Low
012	2023	141.70	High

## Data Availability

All data used for this study have been acquired from data resources publicly available. The observations and ephemerides are available in the RINEX observation and navigation files, respectively. The data are from 4 IGS stations and they can be downloaded from NASA’s Crustal Dynamics Data Information System (CDDIS). Available online: https://cddis.nasa.gov/Data_and_Derived_Products/GNSS/GNSS_data_and_product_archive.html (accessed on 14 January 2023).

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
