# Peer review of "Neustrelitz Total Electron Content Model for Galileo Performance: A Position Domain Analysis"

_sensors, 2023, doi:10.3390/s23073766_

Round 1

Reviewer 1 Report

1. ¿In line 6 "Engeenering" it's properly write?

2. The resolution of figure 5 must be improve.

3. Legend of figure 7 must be declare in page 243

4. The references: 1, 2, 3, 6, 9, 18, 19, 21, 22, 24, 27, 28, 32 could be presented in recently editions or by actuals references (10 years ago maximum)

Author Response

The authors wish to thank the editors and the reviewers for the thorough and useful evaluation of their paper.
The authors have benefited from the reviewer/editor comments and insights and have revised the paper
according to their suggestions.
Detailed replies to the editors/reviewers’ comments are provided in the attached file.

Reviewer 2 Report

Although there are many studies about these 3 models, this study handles the performance analysis from different aspects. 

The article should be revised considering the following situation.

Although anomalies are expected on active days, the selected positions may not have an anomaly on the relevant days. Therefore, choosing a point from an anomaly region with an analysis using a global ionosphere map (it can be any method in the literature) will give more realistic results. This situation should be taken into account.

Author Response

(The authors gave the same response as above.)

Reviewer 3 Report

Dear authors, reading of your manuscript I found three problems:

1) The text was prepared for the submission in very careless manner

2) the research motivation of the paper is very weak and insufficiently substiated.

3) The authors did not managed to tide solar-terrestrial physics and their results clearly and correctly

My recommendation is major revision and re-submission. Hope that my recommendations below will help the authors improving their manuscript.

Author Response

(The authors gave the same response as above.)

Reviewer 4 Report

Revision of: »NTCM-G performance: a position domain analysis«

This article presents the adoption of the NTCM-G model as a trade-off between performance and complexity compared to the earlier Klobuchar and NeQuick-G models. The article is particularly relevant to users of single frequency GNSS receivers, as more and more solar events are now affecting the state of the ionosphere due to the approach of the solar apogee. The main contribution of this study is a detailed comparison of different ionospheric algorithms on the specific ionospheric problem days. The performance of the models is evaluated in the position domain by using only a SPP approach. However, it might be interesting to show also the relative positioning results, at least the DGNSS results during specific time periods. For each model, the authors should present equations and cite the relevant literature.      
Figure 5 regarding the execution times of NTCM-G, NeQuick and Klobuchar should be improved (labelling of the y-axis, more visible and clear labels in the graph).         
When selecting experimental datasets, the authors should reference why they selected those particular days (e.g., previous publications or other measurements, perhaps the I95 index). Although the article is interesting, we would strongly recommend that the authors make it a bit more detailed and compare their observations with other reference data, e.g., relative measurements of the carrier-phase.
Some technical problems in the article, e.g., problematic citation of references (lines 231, 234, 237).

Conclusion: the article is not yet suitable for publication in its present form. The content is interesting, so I encourage the authors to further develop their research and present it in a better form.

Date of this response       
20 March 2023 12:43:00

Author Response

(The authors gave the same response as above.)

Round 2

Reviewer 3 Report

No additional comments

Reviewer 4 Report

Dear Authors,
the improved version of the manuscript is of much higher quality and has the potential for many more readers. However, there are still technical Word issues that you will resolve when you finalise the formatting of the article.

I would like to take this opportunity to congratulate you on the publication of the article.

Regards, Reviewer